# Exploring Differences Between Tabular Enterprise Data and Public Benchmarks

Myung Jun Kim [* 1]   Maximilian Schambach [* 1]   Frank Essenberger [1]   Andre Sres [1]   Johannes Höhne [1]

## Abstract

Tabular data dominate the landscape of data science, increasingly attracting innovative machine learning models and tailored benchmarks. Yet, little is known for enterprise data, where tables constitute the backbone of business operations. To broaden the benchmarking landscape for business applications, this work aims to actualize the characteristics of enterprise data by providing an analysis of data statistics and performance measurements of tabular models such as TabPFN, TabICL and ConTextTab. Through our analysis, we find enterprise data markedly differ from tabular benchmarks and we demonstrate that a tabular model that performs well on typical tabular benchmarks may perform poorly on real world enterprise data – and vice versa. This lack of generalization underlines the need for additional benchmarks with enterprise-grade characteristics.

## 1. Introduction

Tabular data is foundational to enterprise applications, driving decision-makings across finance, healthcare, retail, and logistics (Van Breugel & Van Der Schaar, 2024; Klein & Hoffart, 2025). Despite its importance, tabular benchmarks for real-world applications remain underdeveloped, even as recent advances in tabular foundation models (Grinsztajn et al., 2025; Qu et al., 2026; Spinaci et al., 2025) have reignited interest in the domain and public leaderboards have recently been initiated (Erickson et al., 2025). This gap limits effective evaluation and research progress in addressing real-world tabular prediction problems.

We therefore examine the differences between enterprise tabular tasks and widely-used benchmarks from open-source repositories in a data-driven manner. While open-source datasets are foundational for research, enterprise data reflects richer real-world complexity, domain-specific con-

straints, and often overall larger tables (Bodensohn et al., 2025). Through a large-scale statistical evaluation, we analyze and compare the statistical properties of internally available data from enterprise tasks with popular benchmark datasets, identifying substantial disparities that challenge assumptions about generalizability. Critically, these differences extend to model rankings, with performance and preferences shifting markedly between open-source benchmarks and enterprise tasks. This raises a central question: Do state-of-the-art models optimized on public tabular benchmarks effectively translate to enterprise-grade datasets?

This work pinpoints the limitations of current tabular AI research when applied to real-world enterprise scenarios and thereby demonstrates the need of novel "enterprise-like" benchmarks. Bridging the gap between open-source benchmarks and enterprise data enables tabular AI research to tackle complex real-world challenges that drive the industry.

## 2. Related works

**Tabular benchmarks:** Over the years, numerous public benchmarks for tabular learning of different characteristics have been constructed, but the realm of enterprise-grade datasets remains scarce. OpenML-CC18 (Bischl et al., 2021), OpenML-CTR23 (Fischer et al., 2023), PMLB (Olson et al., 2017), Grinsztajn et al. (2022), TALENT (Liu et al., 2025), and TabArena (Erickson et al., 2025) comprise tables containing mostly of numeric values, where no or little handling of string-entries are required. Meanwhile, CARTE (Kim et al., 2024) and TextTabBench (Mráz et al., 2025) concentrate on tables with semantic meanings (*e.g.*, city names or movie summaries). While more similar to enterprise data, these pertain mostly general knowledge or free-texts, not enterprise-specific semantics. In this regard, TabReD (Rubachev et al., 2025) and SALT (Klein et al., 2025) focus on industrial and business applications. However, TabReD is highly numerical and SALT is limited by the number of datasets.

**Tabular learning:** For many years in tabular learning, gradient boosted decision trees (GBDTs) (Prokhorenkova et al., 2018; Chen & Guestrin, 2016; Ke et al., 2017) have been the forefront. In recent years, however, there has been a series of sophisticated neural architectures that gave significant jumps to match, or even outperform, GBDTs (Holzmüller et al.,

---

[*]Equal contribution [1]SAP SE, Germany. Correspondence to: Myung Jun Kim <myung.jun.kim@sap.com>.

*Proceedings of the $2^{nd}$ ICML Workshop on Foundation Models for Structured Data*, Seoul, South Korea. 2026. Copyright 2026 by the author(s).

2024; Gorishniy et al., 2025). Moreover, tabular in-context learners have spurred the tabular learning community. In particular, TabPFN (Hollmann et al., 2025; Grinsztajn et al., 2025) and TabICL (Qu et al., 2025; 2026) stands strong for numerical-heavy benchmarks while ConTextTab (Spinaci et al., 2025) performs best on semantic-rich benchmarks. Yet, little is known about how these models perform on enterprise-grade datasets.

**Analysis on enterprise data:** Despite the challenge of disclosure, there has been several works that analyze enterprise datasets. Vogelsgesang et al. (2018) explores the differences between enterprise and web data while Kayali et al. (2024) contributes the GOBY benchmark, primarily for table understanding tasks (*e.g.*, column annotation) with LLMs. The work of Bodensohn et al. (2025) dives deeper to the related topic with SAP enterprise data, providing insights and limitations of LLMs for table understanding and database operations tasks – in particular focusing a "column type annotation" task. All these works, however, do not address the tabular learning and there is a need to to further examine real-world enterprise prediction problems and the underlying distinct data characteristics at scale.

## 3. Enterprise data and tabular benchmarks

As an entry point for this study, we sampled 2000 tasks from a vast internal collection of enterprise-grade datasets, covering tabular classification and regression. The datasets were extracted from various lines of business and are directly connected to decision makings in business operations such as manufacturing, sales, or supply chains. We then curated a subset of 557 tasks based on two criteria: (1) Tasks are non-trivial (*i.e.*, a base predictor does not achieve near-perfect score); (2) Tasks contain predictive signal (*i.e.*, a base predictor has at least 20% relative improvement over a naive baseline). We will refer to this collection as the *Enterprise-Grade Internal Benchmark* (`EGI-Bench`).

To compare its statistics, we consider previously discussed benchmarks, consisting of 276 tasks in total: `OS-Tabular` containing six benchmarks commonly discussed for tabular learning – `CARTE`, `OpenML-CC18`, `OpenML-CTR23`, `TabArena`, `TALENT` (tiny), and `TextTabBench`; as well as `OS-Industry` containing two sets of enterprise datasets – `SALT` and `TabReD`. Table 1 shows the task distribution across all benchmarks used.

## 4. Discrepancies in data statistics

To compare the corresponding benchmark statistics at scale, we calculate several statistical measures that cover simple statistical properties, task difficulty, and drift across splits for each individual task. In particular, we measure: (1) the feature data type rates (rate of numerical, string, date,

*Table 1.* Task distribution across all considered benchmarks.

| Source | Classification | Regression | Total |
|---|---|---|---|
| `OS-Tabular` | 152 | 108 | **260** |
| – `CARTE` | 11 | 40 | 51 |
| – `OpenML-CC18` | 68 | 0 | 68 |
| – `OpenML-CTR23` | 0 | 34 | 34 |
| – `TabArena` | 38 | 13 | 51 |
| – `TALENT` | 26 | 10 | 36 |
| – `TextTabBench` | 9 | 11 | 20 |
| `OS-Industry` | 11 | 5 | **16** |
| – `SALT` | 8 | 0 | 8 |
| – `TabReD` | 3 | 5 | 8 |
| `EGI-Bench` | 491 | 66 | **557** |
| **Total** | **654** | **179** | **833** |

and other data types); (2) the number of rows of the train split; (3) the number of features; (4) the feature entropy; (5) for classification tasks the cardinality and imbalance of the target class; (6) for regression tasks the skewness and kurtosis of the target; as well as (7) the covaraite drift. Subsequently, we bin each measure into categories which we refer to as characteristics. For example, measuring the number of rows of each task's train split, we bin the result into Small, Medium, and Large, corresponding to tasks with less than $10\,000$, $100\,000$, or more, respectively. Specific details on all measures and thresholds used for binning are outlined in Appendix A.1.

The results are shown in Figure 1, visualizing the per-benchmark distributions across the investigated data and task characteristics. We observe the following, at times severe, distinct differences between enterprise-grade data and open-source benchmarks:

**Strings are widespread in enterprise data:** In contrast to public benchmarks for tabular learning, the top-left of Figure 1 reveals a clear distinction of strings as the dominant data type for `EGI-Bench`. Such characteristic coincides with findings in previous works (Bodensohn et al., 2025; Kayali et al., 2024; Vogelsgesang et al., 2018): enterprise data is heavily emphasized with a substantial amount of semantically rich texts, categorical data, and codes or IDs.

**Features are more repetitive for enterprise data:** The measured feature entropy (top-right, Figure 1) serves as a proxy for repetitiveness of features: A lower feature entropy suggests lower diversity and thus higher repetitiveness of the feature samples. The results clearly indicate enterprise data showcasing less diversity in the feature space, thus a more dominant "Low" feature entropy distribution as opposed to public benchmarks which show less redundancy.

**Tasks are more complex for enterprise data:** Five statistics in bottom of Figure 1 represent measures associated with the difficulty of the tasks: covariate drift, class cardinality and imbalance for classification, as well as skewness

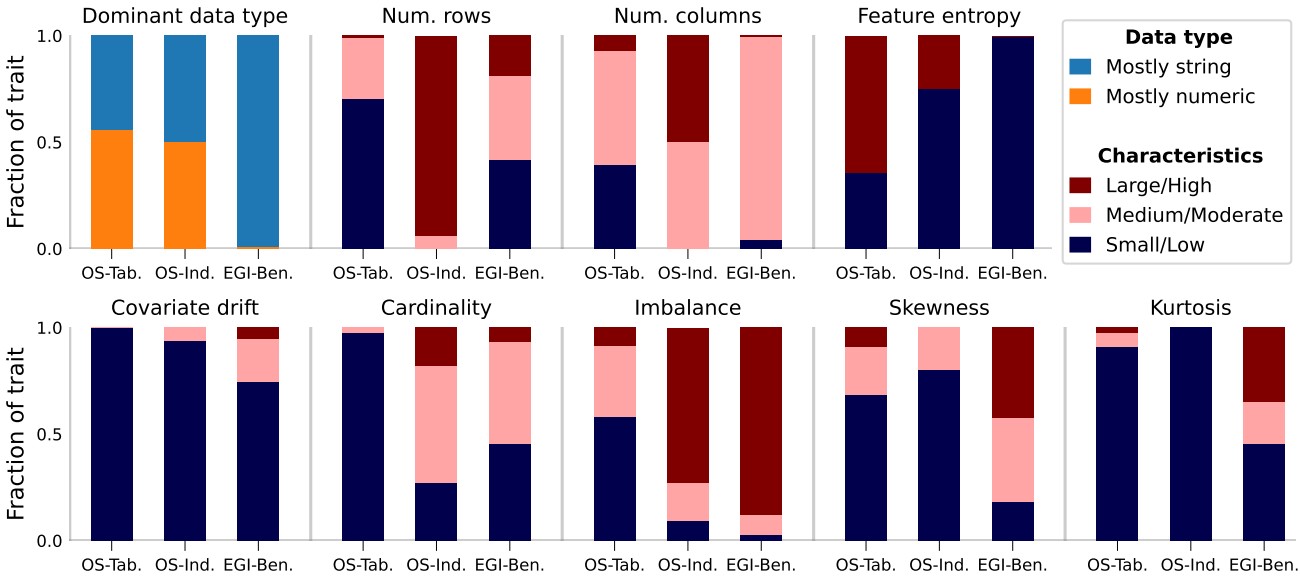

*Figure 1.* Distribution of the evaluated data characteristics across `EGI-Bench` and `OS-Industry` and `OS-Tabular`. The overall comparison shows that for enterprise data: (1) strings are widespread, (2) features are more repetitive, (3) tables are more spread in terms of size, and (3) tasks are more complex, that is having higher cardinality, as well as having more imbalanced or skewed targets.

and kurtosis for regression targets. Compared to public benchmarks, `EGI-Bench` has more prominent covariate drifts. This is typical in data originating from time-bound processes, as previous works indicate (Rubachev et al., 2025). As these drift violate the core IID assumption of many tabular models, they may have significant impact on model performances. Moreover, `EGI-Bench` shows generally higher-cardinality classification targets which are often highly imbalanced. Similarly, for regression, targets are often more skewed and tailed. It is also worth noting that these differences hint the high curation in public benchmarks (*e.g.*, label binarization and target power-transforms in `CARTE`), leading to higher dissimilarity to real-world enterprise data.

Overall, we find a number of distinct characteristics where public benchmarks differ markedly from enterprise-grade tasks as reflected by our investigated `EGI-Bench` suite.

## 5. Discrepancies in model ranks

Given the eminent statistical discrepancies between `EGI-Bench` and public benchmarks, does the relative performance of tabular learners, often optimized on public benchmarks, translate to our `EGI-Bench`? In this section, we aim to answer this mundane but important question.

**Experimental settings:** We measured performances of seven representative tabular learners across `OS-Tabular` and `EGI-Bench`[1]. The baselines include state-of-the art in-context learners (TabPFN-2.5, TabICLv2, ContTextTab),

recent deep learning approaches (RealMLP), as well as conventional per-dataset trained baselines (XGBoost, Random Forest), covering a broad spectrum from classical to modern state-of-the-art models. Detailed information on all evaluated baselines are reported in Appendix A.2. For evaluation, we randomly split each dataset into train and test with ratio of 80:20, respectively (if no split is defined by the benchmark itself). The model performance is measured using balanced accuracy for classification and soft-clipped $R^2$ score for regression. Additionally, we provide ELO score results, following the recent works in TabArena.

**Ranks do not generalize from public benchmarks to enterprise data:** Figure 2 presents the overall results across `OS-Tabular`, and `EGI-Bench` with critical difference diagram[2] (top) and ELO scores (bottom) with scores normalized to Random Forest at 1000.

Notably, the ranks and relative scores are non-consistent across the benchmarks, likely due to the specific statistical characteristics as previously discussed. There is no clear winner for all, and different characteristics of each benchmark favor models tailored for each modality. For example, ContTextTab, with its native handling of string features, performs strongly for `EGI-Bench`, yielding a marked difference compared to other table foundation models (TabPFN-2.5 and TabICLv2). However, it falls slightly behind on numerics-heavy public benchmarks, where TabICL and TabPFN shine. Nevertheless, the state-of-the-art tabular

---

[1]We do not evaluate on `OS-Industry` due to small number of tasks (Table 1).

[2]Ranks on $x-$axis with crossbar representing no statistical difference between the models based on Conover post hoc test after a Friedman pairwise significance test (Conover, 1999).

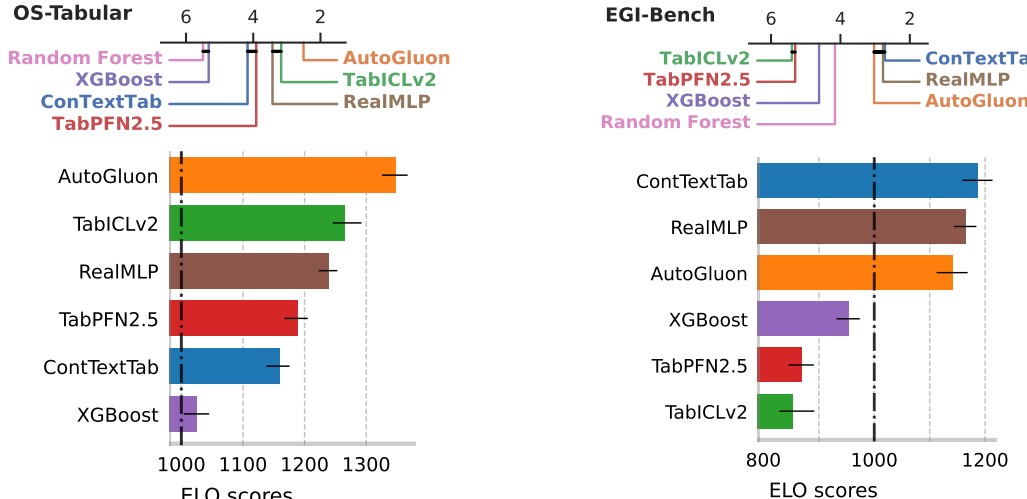

*Figure 2.* **Model performance of tabular learners for public tabular benchmarks does not generalize to enterprise-grade datasets.** Top: Critical difference diagram depicting model ranks and statistical difference between the models; Bottom: ELO scores of tabular learners (with scores normalized to Random Forest at 1000 ELO). Notably, the ranks on `EGI-Bench` markedly differ from those of public benchmarks.

learners, which are optimized for `OS-Tabular` benchmark due to its availability for research-driven iterative improvements, are outperformed even by the classical non-tuned Random Forest on `EGI-Bench`. Only AutoGluon, due to its model-agnostic design and elaborate feature preprocessing, shows a mostly consistent ranking behaviour across the investigated benchmarks. This clearly highlights the challenge of such models to generalize over enterprise data and the overall necessity to broaden the benchmarking landscape beyond numerics-heavy, well balanced, small and low-cardinal public datasets.

## 6. Discussions

**Broadening the tabular benchmark landscape:** Continuous efforts in constructing reliable tabular benchmarks have provided synergistic effect with recent model development as particularly highlighted by the introduction of TabArena (including its leaderboard) and the spurred model development in the recent past. The evaluations on such benchmarks provide instrumental insights that orchestrate effective model development. However, from the perspectives of enterprise applications, where crucial real-world decision makings for business operations reside, the current benchmark landscape is confined to certain data characteristics that are dissimilar to those found in enterprise-grade datasets. In particular, the disparity arises in dominance of strings, more repetitive features, and the complexity and drift of given tasks. Moreover, as we show, these discrepancies can result in difficulties of current tabular learners to generalize over enterprise-data. These shortcomings highlight the need to enlarge the current spectrum of tabular benchmarks to cover these use-cases and data nuances.

**Limitations:** Due to the limited availability of public datasets, our `OS-Industry` benchmark collection is very limited, with only 16 tasks. Moreover, we do not make further distinction within string features, such as free text, categories and IDs which could reveal additional insights. Finally, evaluations are limited to models with default preprocessing and without any customized feature engineering, which could bias results.

**Contextualization is crucial:** In enterprise data, a value rarely carries meaning on its own – it must be interpreted in context beyond its statistical embedding. In an enterprise scenario, *e.g.*, a Material ID like "1000023" is a prime example: for one customer, this may refer to a turbine, while for another, a cardboard box. Its true identity is only unlocked by contextual fields, potentially only available beyond the scope of a single table. This principle extends to numerical fields in transaction data: For example, a "Quantity" column may contain data of different dimensions or units. As such, these values are incomparable without their associated units, as their meaning shifts, *e.g.*, from pieces to tons to hours. Treating such values with a single distribution, as is commonly done in the clean public-data setting, fundamentally misunderstands the data. Even the concept of "missing" may be nuanced, rarely appearing as a clean "NaN" or "NULL". Instead, it might be a default value of the underlying process, such as "00000000" or simply an empty string. However, determining whether a blank is a true omission or a semantically valid state depends again on the context. In essence, a cell's value is just one piece of the puzzle – its true meaning is derived from its semantic context – not just its statistical – and the implicit business process it represents.

# Acknowledgements

We would like to thank Johannes Hoffart, Markus Kohler, and Sam Thelin for their insightful comments and suggestions throughout the development of this work.

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

# A. Additional Details

## A.1. Statistical measures and characteristics

We provide detailed information on the measured statistics and resulting data characteristics below:

**Height:** We measure the number of rows of the training split. The result is binned into Small (below 10k rows), Medium (between 10k and 100k rows), and Large (above).

**Width:** We measure the number of columns/features of the underlying table. The result is binned into Small (below 10 features), Medium (between 11 and 100 features), and Large (above).

**Data type:** We measure the data type of the features of each task. That is, we measure the percentage of numerical (float, int) features, string (string and categorical) features, as well as date and others. If a task has more than 50% numerical features, we consider it as "mostly numerical", if it has more than 50% string feature, as "mostly string".

**Feature entropy:** We estimate the feature entropy via a compression proxy. We do this mainly to obtain an estimate that can be applied to all tables, regardless of the underlying data types and without relying on a featurization on encoding of the table. To this end, we serialize the dataframe column-wise and apply a Zstd compression. We then calculate the average bits per cell of the table, after compression, which we use as the entropy proxy. We then bin the result into Low (below 10) and High (above).

**Classification tasks:** We measure the cardinality of the target and bin it into Low (below 10 classes), Moderate (between 11 and 100 classes), and High (above). In addition, we measure the frequency of each class and calculate the Majority/Minority Ratio (MMR) as the ratio of the probability of the majority class to the median probability across all classes. The higher the ratio, the more imbalanced is the classification target. We bin this into Low (below 1.5), Moderate (between 1.5 and 3), and High (above) to obtain the Imbalance characterstics.

**Regression tasks:** We calculate robust, percentile-based estimated of skew and kurtosis, following the implementations in the `statsmodels` package. We then bin the skew into Low (below 0.25), Moderate (between 0.25 and 0.75), and High (above) and the kurtosis into Low (below 0.75), Moderate (between 0.75 and 1.5), and High (above).

**Covariate drift:** Finally, to estimate the covariate drift, we compute the normalized compression distance of the feature entropy $E$ (estimated via compression as outlined above) on the train and the test split. That is, we calculate

$$E_{\text{NCD}} = \frac{E_{\text{train}} - \min(E_{\text{train}}, E_{\text{test}})}{\max(E_{\text{train}}, E_{\text{test}})} \ . \tag{1}$$

We then bin into Low (below 0.05), Moderate (between 0.05 and 0.1), and High (above).

## A.2. Tabular learning baselines

**AutoGluon** (Erickson et al., 2020): machine learning library that automates data preprocessing, model selection, and hyperparameter tuning with multi-layered ensembling and stacking.

**TabPFN-2.5** (Grinsztajn et al., 2025): a tabular foundation model for in-context learning that leverages prior-fitted-networks. The model is trained with synthetic data. We evaluate TabPFN-2.5 using the official PyPi package version 6.3.2. For classification tasks with more than 10 classes, we use the many-class extension from `tabpfn-extensions` in version 0.2.2 We use its default initialization parameters.

**TabICLv2** (Qu et al., 2026): a tabular foundation model for in-context learning with designated synthetic-data engine and a scalable softmax for longer contexts. The model improves TabICL (Qu et al., 2025) on its pretraining.

**ConTextTab** (Spinaci et al., 2025): a tabular in-context learner with data type-specific encodings, including native string-handling using an `All-MiniLM-6L-v2` langauge model, based on a TabPFNv2-like Transformer backbone. The model has been pre-trained from the real-world T4 dataset Gardner et al. (2024). We evaluate the recent ConTextTab model using the official repository and checkpoints at git version tag v1.1.2[3].

**RealMLP** (Holzmüller et al., 2024): an improved MLP with architectural changes and meta-tuned default hyperparameters, specifically optimized for tabular data. We use the recent 1.7.3 version of the PyPi `pytabkit` package (Holzmüller et al.,

---

[3] https://github.com/SAP-samples/contexttab/

2024) and evaluate RealMLP with 5-fold inner-CV hyperparameter optimization and search spaces from the TabArena setup (Erickson et al., 2025). For string feature preprocessing, we implemented `AutoMLPipelineFeatureGenerator` from AutoGluon (Erickson et al., 2020).

**XGBoost** (Chen & Guestrin, 2016): a representative gradient boosting decision trees learner. We use the implementation from `pytabkit` with the tuned defaults developed from Holzmüller et al. (2024).

**Random Forest** (Pedregosa et al., 2011): A meta estimator that fits a number of decision tree classifiers on various sub-samples of the dataset and averages for improved predictions and overfitting. We use the implementation from scikit-learn version 1.5.2 with default parameters.

