# OpenReview forum: "Exploring Differences Between Tabular Enterprise Data and Public Benchmarks"
_ICML.cc/2026/Workshop/FMSD — FMSD @ ICML 2026 Poster_

### Official Review · Reviewer_QLmk · 2026-05-15
**A Timely Study on the Gap Between Public Tabular Benchmarks and Enterprise Data**

**Rating:** 7
**Confidence:** 4

**Review:**

This paper studies how enterprise tabular datasets differ from commonly used public tabular benchmarks, and how these differences affect the evaluation of tabular learning models. The authors construct an internal enterprise benchmark, EGI-Bench, from a large collection of enterprise-grade classification and regression tasks. They compare its statistical properties with several public benchmark collections, including OpenML-based benchmarks, TALENT, TabArena, CARTE, TextTabBench, SALT, and TabReD. The paper finds that enterprise data differs substantially from public benchmarks in terms of data types, feature repetitiveness, table size, target imbalance, covariate drift, and target distribution properties. The authors further show that model rankings on public benchmarks do not necessarily transfer to enterprise data: models such as TabPFN and TabICL perform strongly on public numerical-heavy benchmarks, while ConTextTab and classical baselines can behave more favorably on enterprise data.

Strengths:

The paper addresses a very relevant and timely question for the workshop. A large part of the recent progress in structured-data foundation models is driven by public benchmarks, but it is not obvious that these benchmarks reflect the properties of real enterprise data. This paper makes that concern concrete and provides useful evidence that the gap is substantial.

I like that the paper does not only compare model performance, but first analyzes dataset characteristics. The discussion around string-heavy features, repetitive enterprise codes, larger and more varied table sizes, covariate drift, high-cardinality targets, and stronger imbalance is useful. These are exactly the kinds of issues that often make real-world structured data different from clean public datasets.

The model comparison is also valuable. The finding that model rankings can change substantially between public benchmarks and enterprise data is important for the community, especially for researchers developing tabular foundation models. It suggests that optimizing too heavily on current public benchmarks may lead to models that are not necessarily well suited for enterprise-grade use cases.

Areas for Improvement:

The biggest limitation is that the most important benchmark, EGI-Bench, is internal and apparently cannot be released. I understand the practical constraints around enterprise data, but this makes the main claims difficult to verify or reproduce. Since the paper’s central contribution depends on this internal benchmark, it would be helpful to provide more details about how the tasks were selected, anonymized, and filtered, and whether any public proxy benchmark could be constructed to reflect similar properties.

The paper is also mostly descriptive. It convincingly shows that enterprise data differs from public benchmarks, but it does not go very far in explaining which specific factors drive the model ranking changes. For example, it would be useful to isolate whether the performance shift is mainly due to string features, high cardinality, covariate drift, class imbalance, larger table size, or some combination of these.

The evaluation protocol could use more clarification. Some models may benefit much more than others from preprocessing, string handling, feature engineering, or hyperparameter tuning. Since enterprise datasets often require careful preprocessing, evaluating models mostly under default settings may not fully reflect their potential in realistic enterprise workflows.

Detailed Comments:
1. The paper would be stronger if it included a more direct attribution analysis of the model ranking differences. For example, the authors could group tasks by dominant data type, target cardinality, imbalance, or drift level, and report model performance within each group. This would help explain why certain models perform better on EGI-Bench.

2. Since EGI-Bench is internal, the authors should provide as much metadata as possible without revealing sensitive information. More details about the business domains, task types, feature types, target definitions, and split construction would make the benchmark more interpretable.

3. The paper mentions that enterprise data is string-heavy, but the current analysis does not distinguish between free text, categorical strings, IDs, codes, and dates stored as strings. These have very different modeling implications. A more fine-grained analysis of string columns would be very useful.

4. The comparison between ConTextTab, TabPFN, and TabICL is especially interesting. It would be helpful to discuss whether ConTextTab’s advantage on EGI-Bench mainly comes from native string handling, pretraining on real-world tables, or other architectural/preprocessing choices.

5. The paper argues that public benchmarks are often too numerical, clean, balanced, and low-cardinality. This is an important point, but the authors could make the contribution more actionable by proposing concrete design principles for future enterprise-like public benchmarks.

6. The evaluation uses random 80:20 splits when no predefined split is available. For enterprise data, temporal or process-based splits may be more realistic, especially given the discussion of covariate drift. The authors should clarify how often the internal tasks use realistic temporal splits versus random splits.

7. AutoGluon appears to show relatively stable behavior across benchmarks. This is an interesting observation and deserves more discussion, since it may suggest that robust preprocessing and ensembling are still very important even in the era of tabular foundation models.

---

### Official Review · Reviewer_itXg · 2026-05-21
**"Rank Reversals" on Enterprise Data vs. Open Source Benchmarks: Would Love To See More Under the Hood, String Handling Not Completely Clear**

**Rating:** 7
**Confidence:** 4

**Review:**

### **Summary**
This paper presents the interesting problem of how open-source tabular datasets (such as those from Kaggle) transfer to real-use-case enterprise data, which rarely makes it into public benchmarks. The authors introduce EGI-Bench, a proprietary benchmark of 557 tasks pulled from live business operations. Their large-scale evaluation and high-level statistical study compare EGI-Bench against both OS-Tabular and OS-Industry public suites. This comparison highlights a critical "rank reversal": state-of-the-art tabular models like TabPFN perform poorly when applied with no pre-processing compared to classical models like XGBoost when moving away from open-source benchmarks. The authors conclude that the most fundamental characteristic differentiating enterprise data from public benchmarks is how heavily it leans into the string data type, alongside having smaller feature entropy and overall more complex task structures.

### **Strengths**
- **Impressive Data Scale:** Curating 557 live enterprise tasks is an impressive effort. This provides a rare, highly valuable look into real-world corporate data that is predominantly locked behind strict privacy walls.
- **Realistic Task Complexity:** EGI-Bench avoids simple problems. By including tasks with high target cardinality, extreme class imbalance, and skewed regressions, it offers a much more realistic test of a model's true utility as shown in Figure 1.
- **Crucial Discussion Insight:** The authors provide deep explanations of why exactly enterprise data breaks standard ML assumptions and motivates future work for all benchmark authors in the tabular field.
- **Perfect Workshop Fit:** This work hits the bullseye of the workshop theme. Demonstrating that state-of-the-art foundation models struggle on real structured data is a highly relevant insight for the community, and yet another signal for curating higher-quality benchmarks which can represent real use cases, so the community stops overoptimizing on play datasets.

### **Areas of Improvement**

- **The "Black Box" Barrier and Data Transparency**: EGI-Bench is proprietary, which is understandable, but maybe too much remains hidden. The paper relies entirely on highly aggregated, binned visualizations (Figure 1). Without an anonymized metadata table or references to structurally similar open-source (OS) datasets, the benchmark remains a black box and it's difficult for the community to build better OS benchmarks with this very limited information.

- **Broad "String" Classification**: The paper's core finding hinges on enterprise data being "mostly string." However, the authors explicitly lump categorical data, IDs, and free-text strings into the exact same bucket without applying any distinguishing heuristics. A feature with 500-word sentences which repeats across the table in 2-3 variations is easily categorizable and handled with tabular models via simple heuristic preprocessing, while a 10-character numerical ID that rarely repeats doesn't hold any generalizable information whatsoever. Making the general "mostly string" label too broad to be acted upon.

- **Unclear Asymmetry in Preprocessing Baselines**: The per-model out-of-the-box preprocessing seems to keep a large bias across the models. While, for example, AutoGluon (and its feature generator for RealMLP) has a well-engineered preprocessing pipeline capable of digesting string features, other models like TabPFN do not come as well-prepared. It is highly unclear how classical models like RF/XGBoost dealt with the string-heavy datasets natively (likely defaulting to dropping non-numerical columns or arbitrary encoding). It is not clear how these models dealt with the string-heavy datasets while many columns may still be purely "string-categorical".

### **Detailed Comments**
**Isolating Architectural Failure vs. Preprocessing Neglect**: The dramatic "rank reversal" of TFMs might be partially due to how default preprocessing maps string/ID columns to arbitrary ordinal integer tokens, which injects severe out-of-distribution noise into their priors. Tree-based models can isolate these arbitrary IDs using step-functions. The authors should clarify if this rank reversal still occurs if a basic, automated text-cleaning pass (such as a bounded discrete dictionary over repetitive strings) is applied first, or if the string features are simply dropped entirely.

**Grounding the Dataset**: To help the community build an accurate mental model of EGI-Bench without breaching corporate privacy, the authors should either provide a randomized/obfuscated sample schema in the Appendix or explicitly point to 1-2 open-source datasets (even if small) that closely mimic the structural distributions they found in their internal data.

**Deeper Profiling of the "String" Modality**: Defining a dataset as "mostly string" simply because 50% of its columns contain text/categoricals limits overall insights about the structure of the datasets. Instead of exclusively relying on feature entropy, the authors could report standard string heuristics (e.g., median string length, unique string cardinality per column, or a histogram showing text length). This would allow readers to differentiate between highly repetitive categorical codes and semantically rich free text.

Minor Note: In the caption for Figure 1, the list format accidentally repeats the number three ("...and (3) tasks are more complex...").

### **Justification of Score**
**Score: Accept**

**Reasoning:** Sourcing and evaluating 557 real corporate datasets is an impressive effort and a highly relevant contribution to the tabular ML community, even if proprietary showing general results shifts from OS benchamarks, making it a perfect fit for this workshop. While the baseline's preprocessing of the string features could be more clearly explained and the dataset transparency leaves a lot to be desired, the sheer realism of the tasks and the clear evidence of model "rank reversal" make this a highly relevant paper. It serves as a vital signal that the community needs to stop overoptimizing on play datasets and start dealing with real contextual bottlenecks. The conceptual strengths of the paper far outweigh its limitations, making it a clear accept.

---

### Official Review · Reviewer_T1st · 2026-05-22
**Review: Exploring Differences Between Tabular Enterprise Data and Public Benchmarks**

**Rating:** 7
**Confidence:** 4

**Review:**

## Summary
The paper compares an internal enterprise-grade tabular benchmark, EGI-Bench, against public tabular benchmarks and shows that enterprise data differs substantially in data types, feature repetitiveness, task complexity, drift, and target distributions. It further evaluates several tabular learners, including TabPFN-2.5, TabICLv2, ConTextTab, AutoGluon, XGBoost, RealMLP, and Random Forest, and finds that model rankings on public benchmarks do not reliably transfer to enterprise tasks. The key claim is that current public tabular benchmarks underrepresent enterprise-relevant characteristics, especially string-heavy and semantically contextual data. The work argues for broader “enterprise-like” benchmarks to better guide development of tabular foundation models.

## Strengths
- The mismatch between public tabular benchmarks and enterprise tabular data is a timely gap in tabular benchmarks and addressing it is a strong contribution.
- The empirical comparison is broad, covering 557 internal enterprise tasks and 276 public benchmark tasks.
- The analysis considers multiple useful dataset characteristics, including data type composition, entropy, covariate drift, class imbalance, cardinality, skewness, and kurtosis.
- The model-rank comparison provides a compelling practical motivation, showing that benchmark performance may not translate to enterprise settings.
- The discussion section offers useful qualitative insight into why enterprise tables require semantic and business-process context.

## Areas for Improvements
- The internal EGI-Bench data cannot be released, which limits reproducibility and makes it difficult to independently validate the central claims. This is an understandable limitation, however, at least some details on the types of tasks would be helpful. Whether the tasks correspond to problems where ML models are already deployed. If the tasks are simply constructed from enterprise databases using the available data, this would be a major limitation because there is a huge difference between the data that is available in enterprise databases and the data that is actually used for predictive tasks.
- More detail is needed on the sampling and filtering process for the 2,000 original enterprise tasks, including possible selection biases. Currently it seems as if nontrivial performance and predictive signal were the only filters. If that alone already filtered the tasks from 2000 to 557 I have strong concerns that the remaining tasks are representative of true enterprise tasks.  To me, this is evidence that the authors simply sampled random targets from databases. If that’s the case the whole analysis may be confounded because the tasks represent enterprise data, but not true enterprise predictive problems.
- The comparison to OS-Industry is limited by only 16 tasks, so conclusions about public “industry” benchmarks should be presented more cautiously. TabRed has a particular focus on temporal data and was not introduced as an enterprise data benchmark.
- The paper would benefit from stronger ablations connecting specific dataset traits, such as string dominance or drift, to specific model failures.
- Some claims about enterprise complexity are plausible but would be stronger with more quantitative breakdowns by business domain or task type.
- Clarification whether the 557 EGI-Bench tasks are independent, or whether multiple prediction tasks may come from related tables or business processes.
- The evaluation uses default preprocessing for most models. Discussing whether this reflects realistic enterprise deployment or disadvantages models that require task-specific feature engineering would help. Since ContextTab is the only model that natively handles text data, this comparison seems unfair.
- The paper should report uncertainty or confidence intervals for key aggregate statistics beyond the visual summaries in Figures 1 and 2.
- The entropy finding could discuss more thoroughly that the enterprise tasks are less curated while the academic benchmarks use datasets that were already prepared to be used for predictive modeling. So naturally, redundant features are often removed.